

# Atherosclerosis, gut microbiome, and exercise in a meta-omics perspective: a literature review

Haotian Tang, Yanqing Huang, Didi Yuan and Junwen Liu

Department of Histology and Embryology, School of Basic Medical Sciences, Central South University, Changsha, Hunan, China

## ABSTRACT

**Background:** Cardiovascular diseases are the leading cause of death worldwide, significantly impacting public health. Atherosclerotic cardiovascular diseases account for the majority of these deaths, with atherosclerosis marking the initial and most critical phase of their pathophysiological progression. There is a complex relationship between atherosclerosis, the gut microbiome's composition and function, and the potential mediating role of exercise. The adaptability of the gut microbiome and the feasibility of exercise interventions present novel opportunities for therapeutic and preventative approaches.

**Methodology:** We conducted a comprehensive literature review using professional databases such as PubMed and Web of Science. This review focuses on the application of meta-omics techniques, particularly metagenomics and metabolomics, in studying the effects of exercise interventions on the gut microbiome and atherosclerosis.

**Results:** Meta-omics technologies offer unparalleled capabilities to explore the intricate connections between exercise, the microbiome, the metabolome, and cardiometabolic health. This review highlights the advancements in metagenomics and metabolomics, their applications in research, and examines how exercise influences the gut microbiome. We delve into the mechanisms connecting these elements from a metabolic perspective. Metagenomics provides insight into changes in microbial strains post-exercise, while metabolomics sheds light on the shifts in metabolites. Together, these approaches offer a comprehensive understanding of how exercise impacts atherosclerosis through specific mechanisms.

**Conclusions:** Exercise significantly influences atherosclerosis, with the gut microbiome serving as a critical intermediary. Meta-omics technology holds substantial promise for investigating the gut microbiome; however, its methodologies require further refinement. Additionally, there is a pressing need for more extensive cohort studies to enhance our comprehension of the connection among these element.

Corresponding author
Junwen Liu, liujunwen@csu.edu.cn

## INTRODUCTION

Cardiovascular diseases (CVDs) have emerged as the leading cause of global mortality and morbidity. In 2016, an estimated 17.9 million lives were lost due to CVDs (*Virani et al., 2020*), with atherosclerotic cardiovascular diseases (ASCVDs) constituting the predominant component. Atherosclerosis (AS), the primary contributor to most ASCVDs, plays a critical role in their onset and progression (*Frostegård, 2013*). Historically, prevailing perspectives linked genetic and environmental factors, such as poor dietary patterns, ambient air pollution, noise, sleep deprivation, and psychological stress, to the development of AS (*Lechner et al., 2020*). Additionally, growing attention has focused on the gut microbiome due to its association with AS (*Witkowski, Weeks & Hazen, 2020*). Given that exercise can affect the gut microbiome, and that the gut microbiome may be associated with AS, exploring the links between the three may provide new insights into the prevention and treatment of AS.

The intricate interplay among a patient's exercise habits, the gut microbiome, and the atherosclerotic process is undeniably complex and diverse, presenting a formidable challenge in terms of comprehension. The formulation of a precise and efficacious exercise prescription designed to enhance a patient's gut microbiome, subsequently paving the way for the attenuation of AS progression, remains a formidable task (*Chen et al., 2018*). Further research of specific types of gut microbiome (*Sumida et al., 2022*), metabolites (*Duttaroy, 2021*) and exercise should be carried out to help build a systematic and generalized understanding.

In this review, we provide a concise overview of metagenomics, which is used for analyzing strain-specific alterations in the gut microbiome, and metabolomics, which focuses on examining disruptions in small-molecule metabolites. Furthermore, we delve into the synergistic application of these two methodologies. Concurrently, employing this integrated approach, researchers are progressively substantiating the role of the gut microbiome as a pivotal intermediary linking physical exercise and AS. We conclude with a summary and perspective on future directions. The primary audience for this review is researchers working on gut microbiome, AS, or exercise interventions, and we hope that our work will inspire the search for new ways to prevent and treat AS.

## METHODS

Our team focuses on advances in the use of meta-omics techniques to study the relationship between the gut microbiome, exercise interventions, and AS. We do this primarily through the PubMed database for literature search. The main keywords of "atherosclerosis", "gut microbiome" and "exercise" were used and combined with the extraction of relevant articles such as "metabolomics", "metagenomics", "intestinal barrier", "NAFLD", "TMA", "SCFAs", "BAs", "LPS", "metabolites", "LDL-C", "immune", along with using "+", "AND", and "OR" for a specific search result. The identified articles were initially checked to determine their appropriateness to the subject, and all the relevant articles were read in detail. We used meta-omics throughout to exclude studies that did not use meta-omics techniques, but 16S rRNA sequencing remained included due to its widespread use. Of the studies in which gut microbiome affected AS, we included studies

that included exercise training and excluded studies that had only dietary or pharmacological interventions and lacked exercise interventions. For the mechanistic review, we searched using specific targets such as "TMAO", "SCFAs" and combined with "atherosclerosis" to include studies with more comprehensive and novel conclusions. Under this strategy, we collected 17 papers related to metabolites, eight papers related to multi-omics co-analysis, 43 papers related to exercise interventions, 13 papers related to inflammation-related microbiology, 16 papers related to atherosclerosis, and 38 papers related to other supplements, for a total of 135 papers, after excluding duplicates and irrelevant literatures, 94 cases that met our objectives were included after reading the abstracts and screening.

During the search process, we did not refine factors such as journal, publication date or journal impact factor. Ultimately, the time span of the references in this review is from 2006 to 2023 (Fig. 1).

## RESULTS

### Meta-omics is gradually becoming a powerful tool

Recent studies have significantly broadened our comprehension of the gut microbiome's involvement in AS. In contrast to conventional analyses that focus solely on the high-level taxonomic composition of the gut microbiome, contemporary approaches demonstrate a heightened interest in discerning the precise composition of the gut microbiome at the resolution of individual genomes. Furthermore, there is a discernible shift in recent research towards surveying microbial gene expression (*Schirmer et al., 2018*), and investigating the metabolites produced by these microorganisms (*Zierer et al., 2018*). This nuanced exploration enables a more granular understanding of the dynamic interactions within the gut microbiome and its consequential impact on the development and progression of AS.

#### *Metabolomics: accurately revealing the mechanism of action of gut microbiome*

Metabolomics, a discipline centered on the comprehensive microbial transcriptome of an organism, facilitates the meticulous observation of subtle changes in biological systems, thereby providing valuable insights into both physiological and pathological mechanisms (*Johnson, Ivanisevic & Siuzdak, 2016*). The field of metabolomics is broadly classified into two principal approaches: targeted and untargeted, contingent upon the specific molecules under examination (*Wang et al., 2020*). The advent of advanced technologies, such as NMR and mass spectrometry, has been pivotal in endowing metabolomics with unparalleled sensitivity (*Marshall & Powers, 2017*). Given the intimate correlation between alternative splicing and metabolic processes, metabolomics holds significant promise in unraveling the intricate mechanisms of AS (*Ussher et al., 2016*). However, despite its potential, metabolomics confronts formidable technical challenges, foremost among them being data analysis. Additionally, the analysis of samples with intricate compositions, such as feces and blood, which encompass both microbiome and host metabolites, poses a significant hurdle. Addressing the imperative of accurately isolating target metabolites is
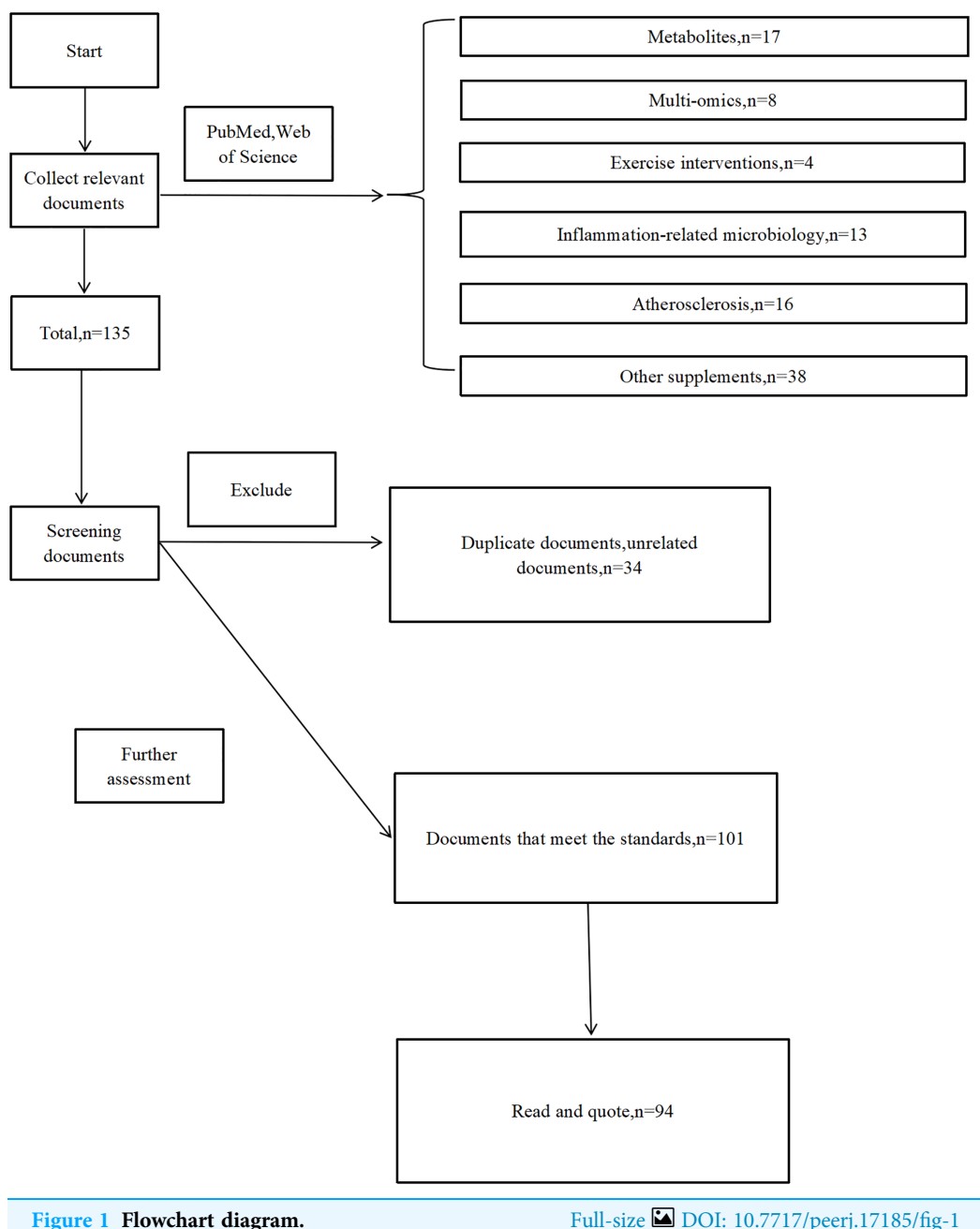

**Figure 1 Flowchart diagram.**

crucial in navigating these challenges (*Khoramipour et al., 2022*). It is imperative to acknowledge, nonetheless, that these challenges, including technical limitations, should not be disregarded. Notably, the ability of metabolomics to construct an intuitive bridge between biological characterization and the realm of small molecule chemicals should not be underestimated (*Bauermeister et al., 2022*).

### 16S rRNA gene sequencing and metagenomics

Historically, gene sequencing necessitated the prior isolation of individual DNA or RNA strands (*Heather & Chain, 2016*). Subsequently, the emergence of culture-independent

sequencing technologies, such as 16S ribosomal RNA (rRNA) gene sequencing, ushered in a more profound understanding of non-cultivated microbiomes (*Knight et al., 2018*). However, it is important to acknowledge that this technology is not without its limitations. One notable constraint lies in the fact that 16S rRNA sequencing is capable of detecting only bacteria and archaea due to the inherent constraints associated with the length of rRNA. Consequently, this method offers a predictive insight into the functional potential of the microbiome rather than furnishing an entirely precise report (*Douglas et al., 2020*). Despite this drawback, 16S rRNA sequencing perseveres as the predominant research method, owing to its convenience and well-established maturity in the scientific community (*Wensel et al., 2022*).

With the progression of technology, the prospect of directly sequencing all genes within a microbial community has materialized (*Ye et al., 2019*). This innovative methodology is commonly referred to as metagenomics. *Sharon & Banfield (2013)* from Berkeley have succinctly defined metagenomics as the scientific discipline that involves "applying modern genomics techniques to directly study communities of microorganisms in their natural state, without the need to isolate single strains in the laboratory". Classical metagenomics typically encompasses four key stages: sample preparation, library preparation, sequencing, and data analysis (*Breitwieser, Lu & Salzberg, 2019*; *Quince et al., 2017*). As biosignature technology advances and sequencing costs decrease, metagenomics is rapidly gaining prominence as the preferred method for microbial community research (*Purushothaman, Meola & Egli, 2022*). This shift is indicative of the method's enhanced efficiency and its capability to capture a holistic view of microbial communities in their natural environments.

### The joint application of meta-omics: the integrated meta-omics

Biochemical reactions in the human body are intricate and multifaceted, and the utility of a single metagenomics approach alone may have limitations. Therefore, the joint application of multiple meta-omics techniques has become increasingly prevalent in scientific research.

We could design different combinations to meet our different research targets. For instance, nonalcoholic fatty liver disease (NAFLD) has correlation with the gut microbiome, in a case-control study, the integrated meta-omics was used to analyse the fecal samples (*Del Chierico et al., 2017*). With the deepening exploration of the gut microbiome and the ongoing maturation of meta-omics methodologies, the collaborative integration of these approaches is poised to become a prevailing trend in the future.

## The tight association between exercise and gut microbiome composition

Previous studies have shown that the gut microbiome composition is determined by the environmental factors more than genetic factors (*Rothschild et al., 2018*). Along with diet, exercise is also an important factor affecting the gut microbiome (*Chen et al., 2018*). Different exercise methods, habits, intensity, *etc.*, will have different impacts on the gut

**Table 1 Exercise intervention affects the gut microbiome as well as atherosclerosis.**

| Author(s), year | Animal model used or subjects' characteristics | Microbiota/ Microbiome analysis method | Exercise intervention | Duration | Key findings |
|---|---|---|---|---|---|
| Patrick C N Rensen, 2022 (*Schönke et al., 2023*) | Atherosclerosis-prone female APOE*3-Leiden. CETP mice | 16S rRNA sequencing, metagenomic shotgun sequencing | Treadmill training in rats 5 times per week for 1 hour each session. Warm-up 15 min at 6–15 m/min and 45 min at 15 m/min; total 839 m per mouse per session | 4 weeks | Late exercise training promotes the enrichment of short-chain fatty acid-producing intestinal bacteria and in turn alleviates atherosclerosis |
| Yu-Tang Tung et al., 2022 (*Huang et al., 2022*) | ApoE knockout mice, wide-type mice | 16S rRNA sequencing, metabolomics | Swimming in a water bath maintained at 35 °C–36 °C for 5 days a week. | 3 months | Endurance exercise mitigates western diet-induced atherosclerosis by ameliorating obesity, inflammatory and chemotactic signals, which are regulated by microbiota and derived SCFAs |
| Zhenqi Zhou et al., 2021 (*Moore et al., 2021*) | Male, whole-body low-density lipoprotein receptor knockout mice | 16S rRNA sequencing | voluntary exercise training | 16 weeks | Low-intensity of active exercise has no significant effect on the gut microbiome and atherosclerosis |
| Nimbe Torres et al., 2019 (*Guevara-Cruz et al., 2019*) | Mexican mestizos between 20 and 65 years of age, with a body mass index (BMI) >18.5 kg/m² and without any chronic disease. | 16S rRNA sequencing | The average number of steps in a week was first quantified. Subsequently, physical activity increased by 10 per cent in the first 15 days, followed by a 25 per cent increase in 1 month and a 50 per cent increase in the second month | 2 months | Exercise intervention significantly reduces the risk of atherosclerosis by decreasing MetS components, small LDL particle concentrations, gut microbiota dysbiosis and metabolic endotoxaemia |
| Monica Aggarwal et al., 2021 (*Ahrens et al., 2021*) | 37 adult males, 36 adult females | 16S rRNA sequencing | 5 hours of dedicated fitness (with three additional hours of sports/ games such as relay races, kick ball, and volleyball) | 6 days | Exercise interventions improved blood lipids and blood pressure and increased the abundance of butyrate producers in the gut microbiome without significant weight loss, suggesting improved cardiovascular health |

microbiome (*Mailing et al., 2019*) (Table 1). The applications of meta-omics methods help us better understand the relationship between them.

### Changes in gut microbiome composition

Researchers once employed 16S rRNA sequencing to compare the fecal samples of athletes with those of ordinary individuals. The study revealed a significant increase in the abundance of gut microbiota in athletes compared to ordinary individuals (*Clarke et al., 2014*). The top six flux changes in relative abundance were in the Firmicutes, *Ruminococcaceae*, S24-7, *Succinivibrionaceae*, *RC9 gut group* and *Succinivibrio* groups (*Clarke et al., 2014*). In another study, which involved fecal sampling, 16S rRNA sequencing, and positron emission tomography, researchers examined twenty-six

sedentary subjects. The findings indicated that exercise has the potential to enhance gut microbiome profiles and mitigate endotoxemia (*Motiani et al., 2020*).

The effect of exercise on gut microbiome is not only limited to the changes of bacterial species, but also the whole intestinal physiology (*Hughes & Holscher, 2021*). The intestinal barrier is a structure that plays a crucial role in maintaining gut homeostasis by regulating the absorption of water, electrolytes, and nutrients from the gut lumen into the bloodstream while preventing the entry of harmful luminal substances and microorganisms (*Julio-Pieper & Bravo, 2016*). Therefore, the intestinal barrier has a huge influence on the composition of the gut microbiome (*Paone & Cani, 2020*). Studies have shown that the intestinal barrier is affected by a variety of factors, including diet and exercise, and the intensity, type and duration of exercise have different effects on it (*Camilleri, 2019*). Moderate exercise improves intestinal barrier permeability, which in turn optimizes the composition of the gut microbiome (*Luo et al., 2014*). This process is complex and not yet fully elucidated. However, exercise is never just more of the same. Excessive exercise can damage the integrity of the intestinal barrier leading to an imbalance in the gut microbiome, which is more pronounced in the absence of adequate rest and nutritional support (*Camilleri, 2019*).

### Meta-omics in the exercise intervention study

Meta-omics techniques are progressively enhancing our capacity to grasp the effects of diverse interventions. Yet, the complex, high-dimensional data these methods produce necessitates the urgent requirement for thorough longitudinal sampling and substantial sample sizes, both of which pose considerable obstacles. Moreover, accurately distinguishing between an individual's fundamental movement capabilities and the precise nature and intensity of their exercise proves to be a daunting task. This underscores the critical need for refined methodologies tailored to individual assessments (*Mohr et al., 2020*).

In research examining the impact of exercise interventions on childhood obesity, scientists utilized metagenomics to pinpoint the specific species and functional capabilities of the gut microbiome. Furthermore, they employed metabolomics techniques to observe metabolic alterations within the gut microbiome that arose due to the exercise interventions. The results demonstrated that these interventions gradually shifted the composition and abundance of the gut microbiome in obese children towards the patterns seen in their normal-weight counterparts. Additionally, significant alterations were observed in metabolites linked to conditions such as obesity and atherosclerosis, including short-chain fatty acids (SCFAs) (*Hu et al., 2022*), have shown changes. The ultimate conclusion is that exercise reduces childhood obesity-induced inflammatory signalling pathways by modulating the microbiome (*Quiroga et al., 2020*). A growing body of research suggests that exercise can influence the gut microbiome with implications for the cardiovascular system. However, identifying the appropriate criteria for exercise and unraveling its specific impacts on both the gut microbiome and the cardiovascular system remains a challenge.

**Table 2  Effects of different types of exercise on atherosclerosis.**

| Author (s), year | Subjects' characteristics | Exercise type | Duration | Key finds |
|---|---|---|---|---|
| Alexandre M. Lehnen et al., 2020 (*Pedralli et al., 2020*) | 42 prehypertensive or hypertensive patients (54 ± 11 years, resting SBP/DBP 137 ± 9/86 ± 6 mmHg) | Aerobic exercise training | 8 weeks | Aerobic exercise significantly improves endothelial function in patients with early hypertension or hypertension and has an important role in delaying atherosclerosis |
| John M Saxton et al., 2018 (*Ashton et al., 2018*) | Non-athlete adults | Resistance training | ≤ 6 weeks, 7–23 weeks, ≥ 24 weeks | Intermediate-term and long-term resistance training reduced systolic and diastolic blood pressure. Intermediate-term resistance training reduced fasting insulin and insulin resistance The effects of resistance training were more pronounced in people with cardiometabolic risk or disease compared to young healthy adults. In conclusion resistance training may improve cardiometabolic health |
| Rune Byrkjeland et al., 2016 (*Byrkjeland et al., 2016*) | Patients with both type 2 diabetes and coronary artery disease (CAD) | Combined training | 150 min per week for 12 months | Combined training has many beneficial effects in patients with type 2 diabetes or CAD, but has limited impact in high-risk, complex patients with type 2 diabetes and CAD combined. Improvements in carotid intima-media thickness were more pronounced with combined training in patients who had not yet developed carotid plaques |

## Effects of exercise and gut microbiome on AS in a meta-omics perspective

Exercise has long been acknowledged as a cornerstone of a healthy lifestyle, demonstrating a crucial role in both preventing and mitigating AS, as previously discussed (*Liu et al., 2023*). We classified exercise into three primary types: aerobic, resistance, and combined exercise, and summarized their impacts on AS in Table 2. As metabolomics, sequencing, and related technologies advance, our grasp of the link between exercise, AS, and the gut microbiome is progressively deepening. Exercise training in humans triggers intricate molecular responses *via* a variety of mechanisms, which go beyond simply altering the composition of the gut microbiome. Metabolomics studies have revealed that changes in the body's metabolites are highly sensitive to environmental influences, physiological shifts, or pathological alterations. As a result, tracking these metabolite changes has become increasingly significant (*Muller, Algavi & Borenstein, 2021*). Metabolites mainly include exogenously ingested compounds (*e.g.*, drugs (*Obach, 2013*), food (*Vernocchi, Del Chierico & Putignani, 2020*)), metabolites of microbial origin, products of microbiome processing exogenously ingested compounds, and host metabolites produced by the body (*Valles-Colomer et al., 2023*). In the human body, these metabolites are usually present as mixtures in different proportions in different samples (*Jang, Chen & Rabinowitz, 2018*). Commonly used samples include faeces, saliva, blood and urine. Through the combined
use of meta-omics, we can deal with different samples and research purposes (*Valles-Colomer et al., 2023*).

### The relationship between exercise and AS in a metabolomics perspective

In the intricate landscape of AS pathogenesis, a complex interplay of factors unfolds. Notably, this process is characterized by discernible fluctuations in metabolites at distinct stages of progression (*Libby, 2021*). Leveraging this hallmark provides us with the opportunity to utilize these pertinent metabolites both as tools for in-depth mechanistic investigations and as discerning markers to distinguish between the various stages of AS (*Wishart, 2016*).

The primary target for intervention in both primary and secondary prevention of ASCVD is the management of low-density lipoprotein cholesterol (LDL-C). Simultaneously, it is essential to recognize that LDL-C functions as a well-established mediator in the development of AS (*Sandesara et al., 2019*). LDL-C is derived from the conversion of very low-density lipoprotein cholesterol (VLDL-C) in the bloodstream. Previous studies consistently demonstrate that lowering LDL-C levels effectively reduces the risk of AS development (*Cannon, 2020*), and LDL-C levels are closely related to exercise training (*Pan et al., 2018*). In conclusion, from a metabolomics perspective, exercise positively impacts the concentration of metabolites closely associated with AS, consequently diminishing the risk of both the onset and progression of this condition. This highlights the potential of incorporating exercise interventions as a preventive strategy against atherosclerosis, complementing the traditional focus on LDL-C management in ASCVD.

### The relationship between gut microbiome and AS in a metabolomics perspective

The gut microbiome produces countless metabolites that exert significant influence on the physiological functions of the human body and the development of diseases (*Heintz-Buschart & Wilmes, 2018*). The types and quantities of metabolites generated by the gut microbiome are highly susceptible to external factors such as diet, exercise, and the body's physiological state. Concurrently, these metabolites exert continuous effects on the body (*Wu et al., 2021*). Of these, the main ones that are more associated with AS are trimethylamine (TMA), SCFAs, secondary bile acids (BAs) (*Witkowski, Weeks & Hazen, 2020*).

TMA is produced by the gut microbiome from foods containing carnitine or choline. After metagenomics analysis, the researchers gradually clarified the main TMA-producing species in the gut microbiome—Clostridium XIVa strains and Eubacterium sp. Strain AB3007 (*Rath et al., 2017*). Subsequently, TMA is oxidized by a monooxygenase containing heparin to form trimethylamine N-oxide (TMAO) in the liver (*Kasahara & Rey, 2019*). Studies have identified TMAO as an independent risk factor for the development of AS. In large cohort studies conducted over extended periods, elevated TMAO levels were significantly associated with cardiovascular issues such as thrombosis (*Zhu et al., 2016*). Moreover, TMAO augments the production of pro-inflammatory
cytokines, such as TNF-α and IL-1β, while attenuating the production of anti-inflammatory cytokines like IL-10. This imbalance in cytokine production exacerbates inflammation and promotes the progression of AS (*Warrier et al., 2015*). In addition, TMAO induces platelet hyperreactivity, which promotes thrombosis, thus causing atherosclerotic thrombotic events.

Other microbial metabolites known for their effects on AS include SCFAs, which are believed to be produced by the gut microbiome through the breakdown of fermentable fibers such as pectin and inulin. Among gut metabolites, SCFAs are considered to be the most abundant, with acetic acid, propionic acid, and butyric acid collectively accounting for about 90% of SCFAs (*Wong et al., 2006*). SCFAs play diverse roles in various biochemical processes in the body and are generally associated with beneficial effects on the cardiovascular system (*Hu et al., 2022*). To investigate the impact of SCFAs on AS, researchers conducted an experiment in which they divided mice into two groups. The experimental group was given drinking water containing propionate, and the results indicated that SCFAs, including propionate, significantly attenuated AS. This suggests a potential protective effect of SCFAs on the cardiovascular system, particularly in the context of AS (*Bartolomaeus et al., 2019*). SCFAs can alleviate AS by acting directly on vascular and renal receptors (*e.g.*, FFAR-2, Olfr78, *etc.*,) (*Le Poul et al., 2003*) and by exerting an anti-inflammatory effect on epithelial cells *via* HDAC (*Li et al., 2018*).

Primary BAs are synthesised by the host in the liver from cholesterol, and later modified by intestinal flora to become secondary BAs (*Staley et al., 2017*). With the application of metagenomics, the species involved in secondary bile acids production in the gut microbiome are gradually being elucidated. Bile salt hydrolase (BSHs) is a key enzyme in this process and is primarily colonized in the Bacteroidetes and Firmicutes phyla in the human gut (*Cai, Sun & Gonzalez, 2022*). Bile acids exert their effects in the body mainly through two receptors: FXR and TGR5 (*Branchereau, Burcelin & Heymes, 2019*). The impact of bile acids on the cardiovascular system has not been fully elucidated, and one of the main reasons for this is that these two receptors often exhibit conflicting effects. *In vitro* studies have shown that FXR agonists upregulate FXR expression in cardiomyocytes and induce their death (*Gao et al., 2017*). However, in some animal experiments, BAs significantly alleviated AS after the activation of FXR receptors (*Vasavan et al., 2018*). This paradox is also reflected in TGR5 (*Pols et al., 2011*), highlighting the intricate regulatory nature of BAs. Consequently, some researchers have identified the BAs axis as central to cardiac metabolism and inflammatory responses, underscoring its complexity (*Guan et al., 2022*).

Moving beyond the three primary metabolites currently under study, it is crucial to recognize the potential significance of other metabolites. Lipopolysaccharides (LPS) stand out as a metabolite closely associated with AS. Within the gut microbiome, Gram-negative bacteria, primarily located in the outer membrane, are major carriers of LPS (*Simpson & Trent, 2019*). Endotoxemia, resulting from the substantial release of LPS into the bloodstream, represents one mechanism through which the gut microbiome adversely affects the organism (*Zhao et al., 2018*). LPS induces a systemic immune response, such as inflammation (*Cani et al., 2008*), while promoting plaque formation in endothelial cells

(*Zhao et al., 2018*), which is associated with AS formation. In metabolomics analysis, researchers have identified a direct correlation between the concentration of high-density lipoprotein (HDL) particles and endotoxemia, with the two exhibiting a negative correlation (*Määttä et al., 2021*). HDL is believed to protect endothelial cells and reduce foam cell formations through various mechanisms, presenting an anti-atherosclerosis effect (*McGillicuddy, Reilly & Rader, 2011*). In a separate NMR metabolomics study focusing on non-lipid or non-lipoprotein metabolites, LPS has been demonstrated to be associated with various cardiometabolism-related compounds, including branched-chain amino acids and aromatic amino acids ('*Hart et al., 2018*). In conclusion, LPS-induced endotoxaemia has a significant contributory effect on AS.

### A metabolomic perspective on the links between exercise, AS, and gut microbiome

The relationship between exercise, the gut microbiome, and AS can be succinctly summarized: both exercise and the gut microbiome can directly impact AS. Additionally, exercise can influence AS by affecting the gut microbiome. These factors are interlinked, and changes in one can directly or indirectly impact the development and progression of AS.

Advancements in metagenomics and metabolomics have made it feasible to target gut microbiome metabolites as a means to alleviate AS by regulating the gut microbiome. Currently, specific treatments for AS are lacking, making lifestyle modifications—such as exercise interventions and dietary changes—the primary therapeutic approaches. Exercise interventions effectively reduce gut inflammation and modulate the gut microbiome, resulting in decreased endotoxaemia and, ultimately, alleviation of AS (*Motiani et al., 2020*). *Battillo & Malin (2023)* conducted a 2-week exercise intervention in obese women and use metabolomics methods to analyse the blood sample. They found that subjects did not experience a significant decrease in blood levels of TMAO, but TMAO decreased more significantly in subjects who originally had high circulating baseline TMAO levels. The researchers hypothesise that this is due to some unknown compensatory mechanism for TMAO in the gut microbiome, but it is worth noting that one of the important precursors for the synthesis of TMAO, choline, showed a significant drop in all the subjects (*Battillo & Malin, 2023*).

Although the specific mechanisms by which SCFAs cause these effects are currently unknown, they still provide mechanistic insights into the treatment of AS by exercise. 16S rRNA sequencing as well as metabolomics analysis of fecal samples from mice after exercise intervention showed a significant increase in the total amount of SCFAs in the cecum (*Li et al., 2023*). This finding reaffirms that exercise can affect the course of AS by influencing the gut microbiome. However, given the large size of the short-chain fatty acid family and the complexity of its mechanism of action, future techniques of multi-omics integration may help us to take our research further.

As with other metabolites, exercise also affects BAs under metabolomics analysis (*Hylemon et al., 2021*), thereby expanding the space for potential therapeutic discovery and development (Fig. 2).

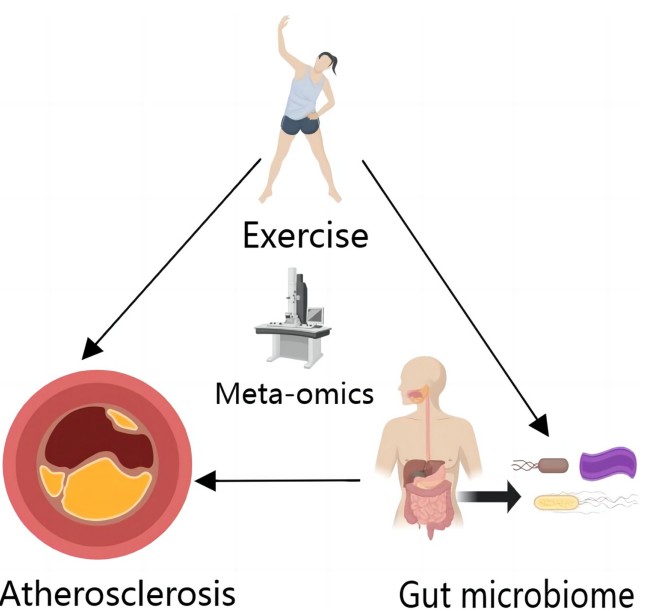

**Figure 2 Exercise and gut microbiome can directly affect AS, and exercise can affect AS by affecting gut microbiome.** In a metagenomics perspective, exercise alters the strain composition of the gut microbiome, which in turn has an impact on AS. In the metabolomics perspective, exercise itself can cause changes in AS-related metabolites such as lipoproteins *in vivo*, and the metabolites of the gut microbiome also contain AS-related metabolites. At the same time, exercise can affect AS by influencing the gut microbiome. This image was created by MedPeer (https://www.medpeer.cn).

### A metagenomics perspective on gut microbiome influencing AS through inflammatory microbiota

Inflammation is a fundamental factor in the development of AS, intricately linked to both the innate and adaptive immune systems (*Wolf & Ley, 2019*). Several signaling pathways associated with the inflammatory response have been implicated in AS, including the NLRP3 inflammasome, toll-like receptors, the proprotein convertase *Bacillus subtilis* protease/kexin type 9, as well as the Notch and Wnt signaling pathways. These signaling cascades play crucial roles in the development and progression of AS (*Kong et al., 2022*). During early life development, the gut microbiome plays a crucial role in regulating the immune system, influencing the differentiation of inflammatory cell types, the production of cytokines, and hematopoiesis (*Dominguez-Bello et al., 2019*). While there have been fewer studies exploring the pro-inflammatory role of gut microbiome in AS, researchers are increasingly recognizing the importance of this connection. For instance, when proinflammatory gut microbiota were transplanted into mice, it led to increased blood leukocyte counts, elevated levels of proinflammatory plasma cytokines, and a higher accumulation of neutrophils in atherosclerotic plaques. Interestingly, this transplantation had no significant effect on plasma lipid levels, TMAO levels, or gut integrity. Consequently, the symptoms of AS worsened in these mice (*Brandsma et al., 2019*). Metagenomics technologies have made characterizing the gut microbiome more accessible. These technologies have revealed that the balance between proinflammatory

and anti-inflammatory microbiota in the gut can vary significantly among individuals, and this difference can also impact AS (*Al Bander et al., 2020*).

Lower-intensity but sustained exercise can effectively reduce inflammation (*Paolucci et al., 2018*), and this process may also be closely linked to the gut microbiome. SCFAs, for example, as described above, have regulatory roles in several signaling pathways in the human body and are primarily anti-inflammatory (*He et al., 2020*) and that SCFAs expressed by the gut microbiome significantly increases after exercise (*Li et al., 2023*). In addition, exercise is effective in relieving chronic systemic inflammation associated with aging (*Angulo et al., 2020*), and given the age-related nature of AS, exercise management is uniquely valuable as a preventive and therapeutic measure for AS. However, prolonged and strenuous exercise can also increase the abundance of proinflammatory microbiota in the gut (*Cataldi et al., 2022*), so it is important to make a reasonable exercise plan.

## DISCUSSIONS

To fully leverage the potential of meta-omics technology, large-scale cohort studies with extensive sample sizes are imperative. Currently, many studies still rely on individual meta-omics techniques, which are clearly insufficient. Integrating multiple meta-omics techniques, such as combining metagenomics and metabolomics, allows us to identify specific strains and elucidate their functions, leading to a more comprehensive understanding of the gut microbiome. Nevertheless, current meta-omics still faces formidable challenges. In metabolomics research, since stool samples, blood samples and other samples with extremely complex compositions (containing both microbiome and host metabolites) are often used (*Krautkramer, Fan & Bäckhed, 2021*), how to accurately and efficiently isolate the target metabolites is the next step that needs to be carefully considered. With the growing sample size, increased depth and resolution of meta-omics, and ongoing advancements in bioinformatics technology, we maintain our belief that meta-omics will play a central role in the future of gut microbiome research.

The gut microbiome serves as the pathway, while the target of our study is AS. Thus, changes in the microbiome are merely the cause. When analyzing the composition of the microbiome, it is essential to take a comprehensive approach, utilizing metagenomics. However, when conducting mechanistic studies, we should avoid getting too entangled in the complexity of the microbiome, which can be challenging to fully describe. Instead, we should swiftly shift our focus to the microbiome's metabolites to delve into molecular mechanisms through compound analysis. Regarding exercise, the challenge lies in accurately controlling variables and categorizing the activities. Therefore, in exercise interventions, researchers often opt for simpler, more isolated movements to facilitate control and computation. However, this approach contradicts the original purpose of exercise intervention, which, as the primary level of treatment and prevention, should be holistic, multifaceted, and life-oriented. Additionally, it is crucial to consider how exercise patterns observed in animal experiments can be translated to benefit humans. Furthermore, the potential dangers of over-exercise are gaining attention, especially when rest and nutritional support are lacking (*Bonomini-Gnutzmann et al., 2022*). How to design an exercise programme wisely, as well as effectively complementing nutritional

strategies, such as changing the proportion of protein in the diet, should also not be overlooked (*Wegierska et al., 2022*). This review offers a meta-omics perspective on how exercise influences the gut microbiome and, consequently, AS. While this perspective is relatively novel, it does not provide a synthesis of intervention methods. Future analyses that integrate exercise with meta-omics, diet, drugs, and other multifaceted are needed.

## CONCLUSION

Exercise serves as a preventive measure, slowing the onset and progression of AS, with the gut microbiome acting as a crucial intermediary between the two. Meta-omics approaches offer substantial potential in elucidating the complex interplay among exercise, the gut microbiome, and AS. However, the concurrent utilization of multiple meta-omics methods remains uncommon. As meta-omics continues to advance and our understanding of exercise and the gut microbiome deepens, there is a promising outlook for the development of effective next-generation AS treatment strategies in the near future.

## ACKNOWLEDGEMENTS

We would like to express our sincere thanks to all the teachers and students of the Department of Histology and Embryology, School of Basic Medical Sciences, Central South University.

### Funding

This research was funded by grants from Hunan Province Natural Science Foundation, No. 2022JJ30780. The funders had no role in study design, data collection and analysis, decision to publish, or preparation of the manuscript.

### Grant Disclosures

The following grant information was disclosed by the authors:
Hunan Province Natural Science Foundation: 2022JJ30780.

### Competing Interests

The authors declare that they have no competing interests.

### Author Contributions

- Haotian Tang conceived and designed the experiments, performed the experiments, analyzed the data, prepared figures and/or tables, authored or reviewed drafts of the article, and approved the final draft.
- Yanqing Huang performed the experiments, analyzed the data, authored or reviewed drafts of the article, and approved the final draft.
- Didi Yuan performed the experiments, analyzed the data, prepared figures and/or tables, and approved the final draft.

- Junwen Liu conceived and designed the experiments, performed the experiments, analyzed the data, prepared figures and/or tables, authored or reviewed drafts of the article, and approved the final draft.

## Data Availability
This is a literature review.

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
