# Peer review of "Atherosclerosis, gut microbiome, and exercise in a meta-omics perspective: a literature review"

_PeerJ, doi:10.7717/peerj.17185_

## Round 0.1 · original submission · Major Revisions

Dear Co-Authors:

Please respond to the reviewers´ comments.

Thank you.

Dr. Manuel Jiménez

**Language Note:** The review process has identified that the English language must be improved. PeerJ can provide language editing services - please contact us at copyediting@peerj.com for pricing (be sure to provide your manuscript number and title). Alternatively, you should make your own arrangements to improve the language quality and provide details in your response letter. – PeerJ Staff

Reviewer 1 ·

Basic reporting

The review manuscript on atherosclerosis, gut microbiome, and exercise lacks a thorough methodological search of the literature and a clear exploration of biological plausibility. A comprehensive review of existing research is essential to establish the context and identify gaps in the current understanding. Additionally, the manuscript should delve into the biological mechanisms linking gut microbiome, exercise, and atherosclerosis, ensuring a robust foundation for the proposed study. Addressing these aspects will enhance the manuscript's overall scientific rigor and contribution to the field.

Experimental design

The study presented lacks a systematic search of the literature, diminishing its methodological robustness. To enhance replicability and facilitate the translation of research findings, the authors should provide more detailed information about their methodology, including a clear description of inclusion and exclusion criteria. A comprehensive and well-documented literature review is crucial for establishing the study's context and ensuring that future researchers can replicate the methodology effectively. Addressing these concerns will strengthen the study's scientific validity and contribute to its impact in the research community.

Validity of the findings

The study's methodology falls short of established standards, making it challenging to determine the internal validity of the findings. Key elements such as article selection, data collection procedures, and data analyses need to adhere to rigorous standards to ensure the reliability of results. Without a robust methodology, the study's overall credibility is compromised, hindering confidence in the reported findings. A thorough revision of the methodology is essential for establishing the study's scientific integrity and contributing to its meaningful impact in the field.

Reviewer 2 ·

Basic reporting

Dear Authors, thank you for your manuscript. In my opinion, the topic is very interesting, but many improvements are needed.
The English you used is not clear and improvements in punctuation and flow are requested.
In my opinion, the beginning paragraphs (1.1, 1.2, 1.3) are well-described. Then, the structure of your review is not very rigorous and professional: the paragraphs are numerated with no logical order (it jumps from 1.3 to 2 and from 2.2 to 3 in the same section) and the manuscript lacks figures and tables. In addition, the main focus of your review did not appear clear in the introduction section, and it is difficult to read the whole work. The structure is confusing, and I could not find a linear thread. In my opinion, you should make your manuscript easier and clearer.

Experimental design

Investigation and methods did not appear rigorous and more effort is needed.
1, a) How many articles did you read?
1, b) How many articles have you selected and included in this study?
A flowchart diagram for article selection is needed.
2) What were the inclusion and/or exclusion criteria for article selection?
Please add a table.
3, a) What is/are the target population/s of your research?
3, b) Are your conclusions valid for men, women, children, older, etc.?
A table with the sample description is needed.
4) Paragraph 2.1-2.2: Exercise is a too-generic term to correlate its effects on AS. What kind of exercise is more effective in improving AS condition? What do you mean by moderate exercise? Maybe you meant moderate-intensity exercise. In my opinion, correction for clarity is necessary. Also, a table that describes how the training parameters could affect AS is needed.

Validity of the findings

In my opinion, at the actual status, this manuscript is not valid for publication. However, with hard effort, I think that you can improve this work.

·

Basic reporting

Please read the attached report.

Experimental design

Please read the attached report.

Validity of the findings

Please read the attached report.

Additional comments

Please read the attached report.

---

## Round 0.2 · accepted · Accept

Dear authors:

After carefully reviewing the reviewers' comments and the improvements made to the final manuscript, I consider that the current version has the conditions for publication in the PeerJ journal. On behalf of the editorial board, I want to thank you for considering PeerJ for the publication of your manuscript.

Greeting.

Dr. Manuel Jimenez

·

Basic reporting

No comment

Experimental design

No comment

Validity of the findings

No comment

Additional comments

The authors made the requested reviews. I have no other comments to do.